# Cutaneous Melanocytic Tumor with *CRTC1::TRIM11* Fusion: Review of the Literature of a Potentially Novel Entity

**DOI:** 10.3390/biology10121286

**Published:** 2021-12-07

**Authors:** Ourania Parra, Konstantinos Linos

**Affiliations:** 1Department of Pathology and Laboratory Medicine, Dartmouth-Hitchcock Medical Center, Lebanon, NH 03756, USA; ourania.parra@hitchcock.org; 2Geisel School of Medicine, Dartmouth College, Hanover, NH 03755, USA

**Keywords:** *CRTC1*, *TRIM1*, melanocytoma, melanocytic differentiation

## Abstract

**Simple Summary:**

Molecular profiling of unclassified neoplasms has been pivotal in the characterization of new entities. Cutaneous melanocytic tumor with *CRTC1::TRIM11* fusion (CMTCT) is a newly described neoplasm that falls into the broad category of diagnostically challenging dermal proliferations with melanocytic differentiation. The aim of this review is to educate colleagues about the clinical, histopathologic, and molecular features of CMTCT, and provide important information on the differential diagnosis.

**Abstract:**

“Cutaneous melanocytic tumor with *CRTC1::TRIM11* fusion” (CMTCT) is a recently described entity belonging to the family of superficial tumors displaying melanocytic differentiation. Thirteen cases have been reported so far, on the head and neck, extremities, and trunk of adults of all ages (12 cases) and one in an 11-year-old child. Histopathologically, it is a nodular or multilobulated tumor composed of spindle and epithelioid cells arranged in nests, fascicles, or bundles that are surrounded by thin collagenous septa. By immunohistochemistry, the tumor shows variable immunoreactivity for S100-protein, SOX10, and MITF, as well as specific melanocytic markers such as MelanA and HMB-45. The neoplasm’s biologic behavior remains uncertain since the reported cases are limited and the follow-up is short (median 12 months). However, local recurrence and synchronous distant metastasis after 13 years of initial resection has been described in one case. Herein, we present a comprehensive literature review of CMTCT hoping to raise awareness among the dermatopathologists of this potentially novel entity.

## 1. Introduction

The histopathologic diagnosis of dermal/subcutaneous proliferations with melanocytic differentiation can be challenging due to existing similarities among different entities and their broad differential diagnosis. This includes the newly described superficial melanocytic tumor with *CRTC1::TRIM11* fusion. Cellier et al. [1] first reported five cases in 2018, and described it as a relatively well-circumscribed neoplasm composed of atypical spindle and epithelioid cells immunoreactive for melanocytic markers, such as SOX10, MITF, and S100-protein; it was initially termed “cutaneous melanocytoma with *CRTC1::TRIM11* fusion”. Since then, eight more cases sharing the same morphologic, immunohistochemical and molecular phenotype have been added to the literature [2,3,4,5]. The histopathologic differential diagnosis includes cutaneous clear cell sarcoma, primary dermal and metastatic melanoma, atypical blue nevus, Spitz tumor, myoepithelial tumor, epithelioid schwannoma, malignant peripheral nerve sheath tumor (MPNST), and paraganglioma-like dermal melanocytic tumor. CMTCT’s biologic behavior has not yet been determined, as the reported median follow-up is relatively short (12 months); one case did recur and metastasized to the regional lymph nodes and lung after 13 years, which prompted the use of the term “cutaneous melanocytic tumor with *CRTC1::TRIM11* fusion” (CMTCT). Herein, we present a review of the current literature on this potentially novel entity.

## 2. Methods

A literature search with the keyword “*CRTC1*::*TRIM11*” was conducted using PubMed database as an online source. Five results were generated, all of which were relevant and included in this review.

## 3. Clinical Features

So far, 13 cases of CMTCT have been reported [1,2,3,4,5], eight on the extremities, three on the head and neck, and two on the lower back. Sizes ranged from 0.4 to 5.1 cm in greatest diameter (median 1 cm), and patient age ranged from 11–82 years (median 33 years). The clinical differential diagnosis included epidermal cyst, pyogenic granuloma, dermatofibroma, and melanoma. The definitive management was local excision in 12 of 13 cases; in one case, the treatment is unknown. Twelve cases have not recurred or metastasized with follow-up ranging from 3 to 72 months (median 12 months). However, one case exhibited local recurrence and synchronous regional and distant metastasis 13 years after initial resection. Information on the primary lesion is not available.

## 4. Histopathologic and Immunohistochemical Features

CMTCT most commonly presents as a well-circumscribed dermal (or dermal and subcutaneous) nodular or multilobulated tumor, with pushing borders, occasionally surrounded by a fibrous capsule. Three cases have exhibited a focally infiltrating pattern of growth [1,3]. A polypoid appearance has also been reported [4]. The neoplasm may abut and/or attenuate the overlying epidermis; however, an intraepidermal component has not been described so far. The neoplasm is composed of nests, fascicles, bundles, or sheets of mostly non-pigmented spindle and epithelioid cells, surrounded by thin collagenous septa. Multinucleated giant cells may also be identified. Cytologically, it exhibits pale to eosinophilic cytoplasm, mild to severe pleomorphism with mild to moderate nuclear atypia, and prominent nucleoli and nuclear pseudoinclusions. All reported cases [1,2,3,4,5] displayed mitotic activity, ranging from 1 to 12 mitoses/10HPF. Areas of necrosis have also been identified in three cases so far [4]. The tumor may be accompanied by a peripheral lymphocytic infiltrate. The histopathologic features of a CMTCT case are presented in Figure 1.

Immunohistochemical features of CMTCT include consistent diffuse SOX10 and MITF immunoreaction [1,2,3,4,5]. MelanA and HMB-45 expression may be focal or patchy and, in some instances, completely absent. Most of the cases reported were also diffusely S100-protein positive; however, in a few cases, S100-protein expression was patchy, focal, or absent. INI1 nuclear expression was retained in all cases tested so far [3], and p16 has been reported as focally positive in a sole case [4]. In total, six cases have also been tested positive for NTRK1 immunohistochemistry (IHC), but the FISH studies failed to identify *NTRK1* fusions or amplifications [1,5]. Lastly, TRIM11 IHC was positive in all five cases examined [1]. Various immunohistochemical stains that have been reported negative are SMA, desmin, caldesmon, CD163, pankeratin, calponin, CD68, CD34, p63, WT1, synaptophysin, chromogranin, CD99, NFP, Neu-N, GFAP, ALK, and ROS. The immunohistochemical profile of a CMTCT case are displayed in Figure 2A–D.

## 5. Molecular Diagnostics

The *CRTC1::TRIM11* fusion has been detected in various studies by RNA sequencing and RT-PCR/direct sequencing, as well as fluorescence in situ hybridization (FISH) (Figure 2E) [1,2,3,4,5]. Chromogenic in situ hybridization and FISH has also been utilized to detect *TRIM11* rearrangements [3]. *EWSR1* break-apart FISH was performed in a subset of cases to exclude clear cell sarcoma. In two cases array-comparative genomic hybridization (CGH) detected gain of the whole chromosome 7, and one case was negative for cytogenetic abnormalities [1,5]. The detected *CRTC1::TRIM11* fusion transcript in the cases examined involved exon 1 of *CRTC1* and exons 2–6 of *TRIM11* [t(19;1)(p13.11;q42.13)]. The exact mechanism of action of the fusion product remains to be elucidated.

## 6. Molecular Pathogenesis

CREB-regulated transcription coactivator 1 (*CRTC1*) is a gene located on the short arm of chromosome 19, and encodes for a protein that activates the cAMP response element-binding (CREB) protein, which is involved in the gene expression that regulates cell proliferation and differentiation [6]. Physiologically, *CRTC1* expression is limited to a few tissues, such as the brain, where it is crucial in hippocampal-dependent memory, neuronal plasticity, dendritic growth, the suprachiasmatic circadian clock, and central nervous system satiety network. It is also involved in metabolism control [7]. In addition to its normal functions, *CRTC1* and its isoforms also promote tumorigenesis. Schumacher et al. [7] showed that *CRTC1* activation plays an important role in colonic adenocarcinoma growth, through the enhancement of prostaglandin E2 function, which includes *NR4A2*, *COX2*, *AREG,* and *IL-6* expression. It is also implicated in the pathogenesis of multiple neoplasms as a fusion partner. The *CRTC1::MAML2* fusion has been detected in mucoepidermoid carcinomas of the salivary gland [8], breast [9], and liver [10], as well as clear cell hidradenomas [11]. The *CRTC1::MAML2* fusion product upregulates CREB induced transcription, leading to oncogenesis; however, Chen et al. [12] described a non CREB-mediated transcription stimulation, which includes interaction with other transcription regulators, including MYC, TP53, NF-κB, ATF2, GLI1, STAT6, and AP-1. A *CRTC1::SS18* fusion has also been reported in a subset of undifferentiated small round blue cell sarcomas [13].

A total of six cases of CMTCT displayed TrkA expression by immunohistochemistry, without corresponding *NTRK1* gene fusion or amplification. The exact mechanism of the protein overexpression has not been studied in CMTCT; it may reflect true over-transcription of the gene through other mechanisms that may be related to the CRTC1::TRIM11 chimeric protein activity [5]. Increased NTRK1 mRNA and corresponding TrkA protein overexpression have also been described in two cases of undifferentiated small round blue cell sarcoma with *CRTC1::SS18* fusion. This finding further supports the possible correlation between *CRTC1* rearrangements and aberrant NTRK1 expression [13]. The overexpression of a Trk protein without corresponding gene amplification or fusion has also been described in other tumors. Kao et al. [14] reported upregulation of *NTRK3* mRNA with corresponding TrkC protein overexpression detected by Pan-TRK immunohistochemical antibody in the majority of *BCOR* and *YWHAE* rearranged sarcomas studied.

Tripartite motif-containing protein 11 (*TRIM11*) is a gene located on the 1q42.13 region, and encodes for a protein with E3 ubiquitin ligase activity that participates in protein degradation in various tissue types [15]. Increased TRIM11 mRNA expression-compared to normal tissue- has been described in hepatocellular carcinoma, which translated to increased protein expression by IHC [16], lung non-small cell carcinoma (NSCC) [15], prostatic, colonic, gastric adenocarcinomas [17,18,19], and lymphoma [20]. *TRIM11* upregulation is also associated with increased motility, invasiveness, and cell proliferation of lung and breast cancer cells and glioblastoma cell lines [15,21,22]. Conversely, when *TRIM11* was knocked down in ovarian cancer cells and lymphoma tissues/lymphoma cell lines in vitro, suppression of cell proliferation was observed. This was the result of either an increase in apoptotic factors or the prevention of cell cycle progression to S or G2 phase [20,23]. In lymphoma tissue and cell lines in particular, the knockdown of TRIM11 resulted in decreased expression of β-catenin, Cyclin D1, and c-Myc, while Axin1 was increased due to decreased ubiquitin-induced degradation. [20] Furthermore, Huang et al. [24] observed increased CD31 expression and microvascular density in lung adenocarcinoma cell lines with upregulated *TRIM11* compared to *TRIM11* downregulated cell lines, indicating stimulation of angiogenesis. High TRIM11 expression was also recognized as an independent marker of poor prognosis in patients with breast cancer, colon and prostate adenocarcinoma, lung NSCC, HCC, and glioblastoma and it has also been correlated with advanced disease stage. [15,16,17,18,20,21,22] Although TRIM11 expression by IHC has been examined in a variety of neoplasms, data on entities that lie within the differential of *CRTC1::TRIM11* melanocytic tumor are limited. Hence, more studies to assess its sensitivity and specificity in identifying CMTCT are necessary.

The reported cases’ clinical, histopathologic, immunohistochemical and molecular features are presented in Table 1.

## 7. Differential Diagnosis

*Cutaneous clear cell sarcoma* (*CCS*): The main differential diagnosis of CMTCT is primary or metastatic cutaneous clear cell sarcoma (CCS). Similar to CMTCT, primary superficial CCS typically presents as a single dermal/subcutaneous nodule or multilobulated tumor organized in nests or fascicles of spindle and epithelioid cells with pale to eosinophilic cytoplasm, surrounded by delicate collagen fibers [25]. It involves the dermis and possibly the subcutis in an infiltrative manner, and can also rarely display epidermotropism, with isolated tumoral nests at the dermo-epidermal junction [26]. In the majority of cases, wreath-like giant cells, which are different from the multinucleated giant cells of CMTCT in terms of the distribution of the nuclei, are present. Immunohistochemically, CCS displays positivity for S100-protein, HMB-45, MelanA and SOX10. The pathogenic driver event of CCS is rearrangement of the *EWSR1* gene with *ATF1* (90% of the cases), *CREB1* or *CREM* as fusion partners [27,28]. Compared to CCS, CMTCT displays mostly a pushing border rather than an infiltrating one and does not involve the epidermis. Furthermore, CMTCT appears to behave in a less aggressive fashion than CCS; hence distinction between these two entities, possibly with the input of molecular techniques, is imperative. Based on their case of recurrent and metastasizing CMTCT, Bontoux et al. [2] argued that this entity represents a cutaneous CCS with a fusion other than the most common *EWSR1::ATF1*. Currently, there is no evidence that the two neoplasms share similar pathways of pathogenesis, and a definitive classification of CMTCT would be premature. More studies with emphasis on the activity of the *CRTC1::TRIM11* fusion product are needed to further classify CMTCT.

*Other melanocytic neoplasms*: Primary or metastatic melanoma confined to the dermis/subcutis lies within the differential and has been the initial diagnosis in 5 cases of CMTCT. [1] Primary dermal melanoma (PDM) also presents as a single nodule or multilobulated tumor composed of spindle, or epithelioid cells arranged in sheets or nests [29,30]. Since a subset of PDMs behave in a low grade fashion compared to similar thickness conventional melanomas, it is plausible that some of them may represent CMTCT. [1] The detection of *CRTC1::TRIM11* fusion and/or the lack of characteristic mutations and cytogenetic abnormalities of PDM could aid in the distinction between the two entities [4,31,32,33]. To exclude a metastatic or regressing melanoma, an extensive histopathologic and clinical work-up are necessary. TRIM11 immunohistochemistry may also have some value as a screening tool for CMTCT; however, TRIM11 has not been examined in melanoma and other dermal proliferations with melanocytic differentiation. More studies are necessary to determine the marker’s diagnostic utility. [1] Albeit rare, NTRK-fused metastatic Spitz melanomas with corresponding anti-Trk immunoreaction have been reported in the literature [34]. Therefore, Trk immunohistochemistry may not be useful in the differential diagnosis.

Spitz tumors also display spindled and/or epithelioid cytological features; however, the majority of them have a different pathway of pathogenesis with fusions in oncogenic kinase drivers, such as *ROS1*, *ALK*, *NTRK1*, *NTRK3*, *MAP3K8*, *MET*, *BRAF*, and *RET* [35,36]. As mentioned above CMTCT expresses TrkA by IHC, however, *NTRK1* rearrangements have not been described. Spitz tumors with *NTRK* fusions display distinct histologic features that may aid in the differential diagnosis. They are usually compound, with epidermal hyperplasia and thin rete ridges [37]. Approximately half of them display a wedge-shaped architecture, and the majority of them is composed of small spindle and/or epithelioid cells that can form rosette-like structures [35,38].

*Clear cell tumor with melanocytic differentiation and MITF rearrangements:* Recently, de la Fouchardiere et al. [39] reported seven cases of an intradermal tumor with clear cell features and expression of melanocytic markers, harboring the *ACTIN* (*ACTG1* or *ACTB*)*::MITF* fusion. Similar to CMTCT, as well as to CSS, the neoplasm presented as a dermal nodule, occasionally involving the subcutis, in patients of all ages. The lesional cells mostly displayed a clear-appearing cytoplasm, and were arranged in small nests or cords; however, they were typically more confluent. The degree of atypia and mitotic activity was also similar to those of CMTCT. MelanA, HMB-45, MITF, S100-protein, and SOX10 immunoreaction confirmed the melanocytic differentiation of this tumor. Although *MITF* hyperactivity could explain the morphologic and immunohistochemical phenotype of this neoplasm, the exact mechanism of action of the *ACTIN::MITF* chimera is still unknown. De la Fouchardiere et al. [40] also reported a case of clear cell tumor with melanocytic differentiation (diffuse SOX10, MITF and S100-protein positivity), harboring *MITF::CREM* fusion, and resembling clear cell sarcoma. Compared to CMTCT, the neoplasm displayed a higher degree of atypia, an infiltrating pattern of growth, and perineural invasion.

*Other soft tissue neoplasms*: Myoepithelial tumors display epithelioid or spindle cytology with S100-protein and SOX10 immunoreactivity. However, in contrast to CMTCT they also display immunoreaction for (myo)epithelial markers such as EMA, low-molecular weight cytokeratins, calponin, and p63 and a subset of them harbors *EWSR1* rearrangements [41]. Epithelioid schwannoma and epithelioid malignant peripheral nerve sheath tumor (MPNST) lie in the differential as well, but they are consistently negative for melanocytic markers including MITF, and a subset of them shows loss of INI1 expression [42,43].

*Paraganglioma-like dermal melanocytic tumor (PDMT*): Paraganglioma-like dermal melanocytic tumor is an entity that was first reported by Deyrup et al. in 2004 [44]. Similar to CMTCT, PDMT is described as a relatively well-circumscribed nodule or multinodular tumor confined to the dermis and/or subcutis with epithelioid to spindle cells arranged in nests, packets, or cords, and surrounded by fibrous septa. A subset of cases exhibits an infiltrating growth pattern. By immunohistochemistry, PDMT is positive for S-100 protein, MITF and HMB-45, variably positive for MelanA and negative for cytokeratins, EMA, SMA, CD34, and synaptophysin. Given the fact that data on the molecular profile of PDMT are not available and both entities display a similar biologic behavior, it is conceivable that a subset of them may harbor the *CRTC1::TRIM11* fusion.

## 8. Conclusions

CMTCT is a potentially novel entity that expands the differential diagnosis of superficial tumors with melanocytic differentiation. Molecular studies may be necessary to reach a definitive diagnosis, especially in challenging cases, when neoplasms with a more aggressive biologic behavior (CCS, melanoma) cannot be categorically excluded. Current data suggest that CMTCT behaves in a low-grade fashion, with only one reported case of recurrence and metastasis after 13 years; however, longer clinical follow-up and additional cases are necessary for definitive conclusions. Treatment of CMTCT was wide excision in the majority of the cases. Targeted therapy with NTRK inhibitors has proven extremely beneficial for patients with NTRK-rearranged neoplasms. Nonetheless, whether or not NTRK upregulation without the corresponding fusion, as observed in CMTCT, is therapeutically actionable remains to be studied. Lastly, although, the role of *CRTC1* and *TRIM11* in tumorigenesis is well-described, the exact mechanism of action of their fusion product and subsequently the definitive classification of this neoplasm has not yet been determined.

## Figures and Tables

**Figure 1 biology-10-01286-f001:**
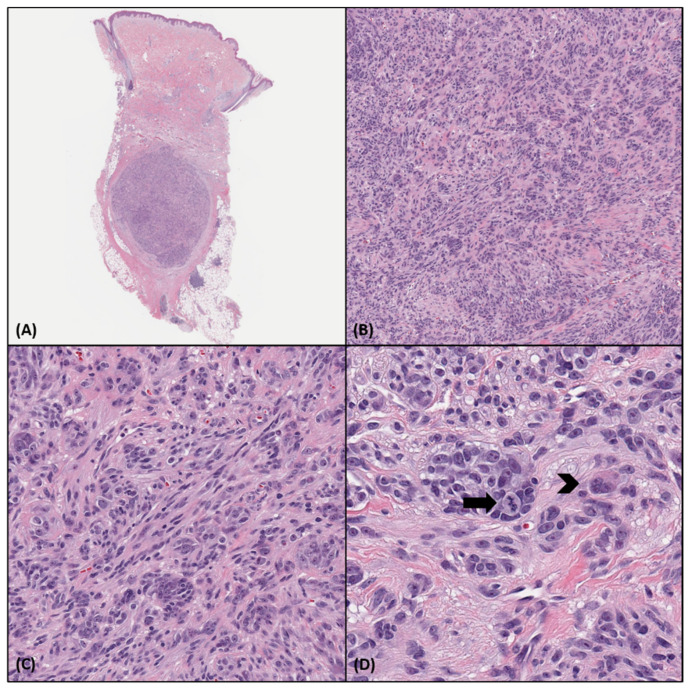
Histopathologic characteristics of a CMTCT case. (**A**). Punch biopsy specimen showing a well-circumscribed encapsulated nodule in the deep dermis and subcutis. (H&E, 10×) (**B**,**C**). The tumor is predominantly arranged in nests and bundles that are surrounded by thin collagenous fibers. It is composed of relatively uniform epithelioid and spindle cells with eosinophilic cytoplasm and round to oval nuclei with prominent nucleoli. (H&E, 100×, 200×) (**D**). A mitotic figure (black arrow) and a multinucleated cell (black arrowhead) are noted. (H&E, 400×).

**Figure 2 biology-10-01286-f002:**
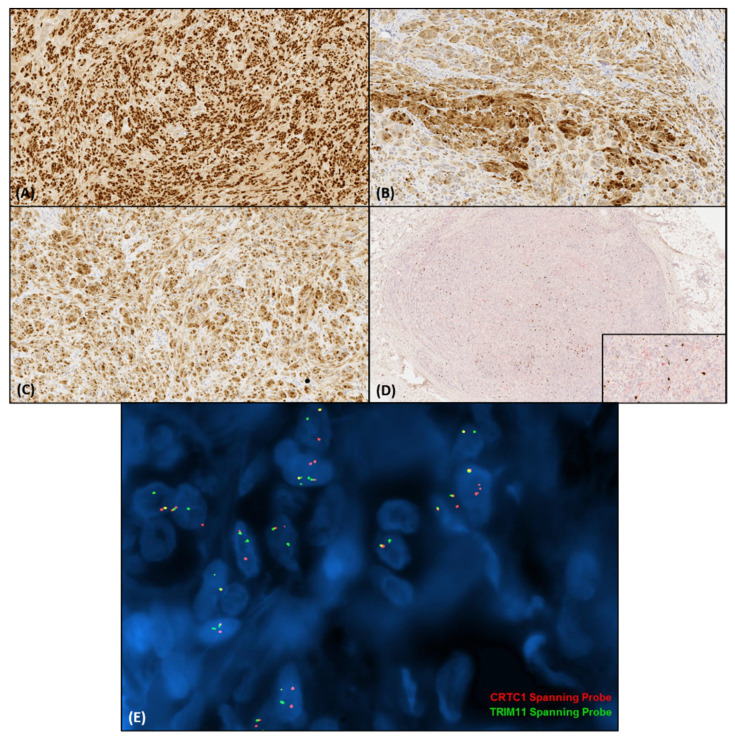
Immunohistochemical features of CMTCT. (**A**). SOX10 shows diffuse nuclear immunoreaction (140×). (**B**). S100-protein shows nuclear and cytoplasmic positivity (140×). (**C**). MITF is also diffusely positive (140×). (**D**). The double staining of MelanA and Ki-67 shows a low proliferation index (20×) and focal weak MelanA immunoreaction (inset, 100×). (**E**). Fluorescence in situ hybridization revealed the juxtaposition of the red (*CRTC1*) and green (*TRIM11*) signals. (Study kindly performed by Julia A. Bridge, University of Nebraska Medical Center).

**Table 1 biology-10-01286-t001:** Clinical, histopathologic, immunohistochemical and molecular characteristics of all published CMTCT cases.

Author	Case	Age	Sex	Location	SOX-10	MITF	S100	MelanA	HMB-45	TRIM11	NTRK1	*CRTC1*:: *TRIM11* Fusion (RNA Sequencing)	TRIM11 FISH	NTRK1 FISH	CGH	Recurrence	Follow-Up (Months)
Cellier et al. [1]	1	28	F	Leg	+	+	+	+ ^a^	+ ^d^	+	+	+	+	−	+7	No	36
	2	82	M	Lumbar	+	+	+	+ ^a^	+ ^a^	+	+	+	+	−	−	No	6
	3	25	F	Elbow	+ ^a^	+	+	+	−	NP	+	+	+	−	NP	No	14
	4	28	F	Thigh	+	+	+	+	+ ^d^	+	+	+	+	NP	+7	No	72
	5	64	M	Neck	+	+	+	−	+ ^d^	+	+	+	+	NP	NP	No	3
Bontoux et al. [2]	6	31	F	Arm	+	NP	+	+	+ ^a^	NP	NP	+	NP	NP	NP	Yes ^e^	156
Kashima et al. [3]	7	77	M	Thigh	+	+	+	+ ^b^	+ ^a^	+	NP	+	CISH +	NP	NP	No	12
Ko et al. [4]	8	32	M	Ear lobe	+	NP	+ ^a^	−	−	NP	NP	NP	+	NP	NP	No	16
	9	59	F	Face	+	NP	−	−	−	NP	NP	NP	+	NP	NP	No	12
	10	11	F	Lower leg	+	NP	−	+ ^a^	+ ^a^	NP	NP	+	+	NP	NP	No	10
	11	49	M	Leg	+	NP	+ ^b^	−	−	NP	NP	+	+	NP	NP	No	10
Parra et al. [5]	12	65	F	Back	+	+	+	+ ^c^	+ ^c^	NP	+	+	−	−	−	No	48
	13	33	F	Bicep	+	NP	+	+ ^b^	NP	NP	NP	+	+	NP	NP	No	9

F, Female; M, Male; NP, Not Performed; FISH, Fluorescence in situ Hybridization; CGH, Comparative Genomic Hybridization; CISH, Chromogenic in situ Hybridization. ^a^ Focal positivity. ^b^ Patchy positivity. ^c^ Rare positivity. ^d^ Positive in a few cells. ^e^ Local recurrence, lymph node and distant metastasis.

## Data Availability

No new data were created or analyzed in this study. Data sharing is not applicable to this article.

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
