# Peer review of "Cutaneous Melanocytic Tumor with CRTC1::TRIM11 Fusion: Review of the Literature of a Potentially Novel Entity"

_biology, 2021, doi:10.3390/biology10121286_

Round 1

Reviewer 1 Report

Greatings for the paper!

I suggest to change the following points:

-In Abstract: line 8, before the bracket, there is only one <">.

- In general all the citation are inserted after the point and not before.

- Explain better the "clinical featurres" and "Conclusion"

Author Response

Reviewer 1.

  • In Abstract: line 8, before the bracket, there is only one <">.

Author response: Brackets have been added.

  • In general all the citation are inserted after the point and not before.

Author response: The citation is inserted before the period when it does not refer to the entire sentence. For instance, citation 20 in this sentence “which translated to increased protein expression by IHC [16], lung non-small cell carcinoma (NSCC) [15], prostatic, colonic, gastric adenocarcinomas [17-19] and lymphoma [20]” only refers to lymphoma.

  • Explain better the "clinical features" and "Conclusion"

Author response: Thank you for your comment. We have included all the described clinical features from the relevant studies. In the conclusion, we summarize the most important clinical and histopathologic features of this entity with emphasis to the molecular implications. We do not have any additional comments.

Reviewer 2 Report

This is a well organized manuscript and worthy for publication. 

Author Response

  • This is a well-organized manuscript and worthy for publication.

Author response: Thank you.

Reviewer 3 Report

A well-performed systematic review exploring the possibility of a new entity in dermatopathology, a tumor with melanocytic differentiation showing CRTC1-TRIM11 . Given the very low number of case reports in this literature, this paper may be useful to clinicians and may be eligible to be published after minor revisions; I have only minor queries:

CRTC1-TRIM11 fusion (RNA sequencing) was not performed in 2 of 4 patients of the Ko study... why did you decide to include them in the review?

A materials and methods section saying what keywords were used and what databases were searched should be added

Author Response

  • A well-performed systematic review exploring the possibility of a new entity in dermatopathology, a tumor with melanocytic differentiation showing CRTC1-TRIM11. Given the very low number of case reports in this literature, this paper may be useful to clinicians and may be eligible to be published after minor revisions.

Author response: Thank you for your comments.

  • CRTC1-TRIM11 fusion (RNA sequencing) was not performed in 2 of 4 patients of the Ko study... why did you decide to include them in the review?

Author response: We decided to include them because the TRIM11 rearrangement was confirmed by Fluorescence in situ hybridization.

  • A materials and methods section saying what keywords were used and what databases were searched should be added.

Author response: A “Methods” section was added to the manuscript. “A literature search using the keyword “CRTC1-TRIM11” was conducted using PubMed as an online source. Five results were generated, all of which were relevant and included in the literature review.”

Reviewer 4 Report

A manuscript by Parra and Linos comprehensively discusses the current knowledge on CRTC1-TRIM11 fusion in cutaneous melanocytic tumor. A review is well and clearly written, references are appropriate. The novelty of the manuscript is high.

Author Response

  • A manuscript by Parra and Linos comprehensively discusses the current knowledge on CRTC1-TRIM11 fusion in cutaneous melanocytic tumor. A review is well and clearly written, references are appropriate. The novelty of the manuscript is high.

Author response: Thank you.

Reviewer 5 Report

This review article entitled “Cutaneous melanocytic tumor with CRTC1-TRIM11 fusion: Review of the literature of a potentially novel entity” by Ourania Parra and Konstantinos Linos is a comprehensive review about cutaneous melanocytic tumor characterized by CRTC1-TRIM11 fusion gene. The descriptions on this paper are correctly based on the previous papers. The beautiful figures and well-organized tables on the current form will help readers understand the contents easily and precisely. This paper will be of interest to readers of this journal. I have only one concern mentioned below.

Comment

I am interested in macroscopic appearance of this tumor because it is one of the most important findings when clinicians inspect the skin of a patient. Are there any characteristic gross findings with this tumor? The authors are encouraged to add this information in Clinical features.

Author Response

  • This review article entitled “Cutaneous melanocytic tumor with CRTC1-TRIM11 fusion: Review of the literature of a potentially novel entity” by Ourania Parra and Konstantinos Linos is a comprehensive review about cutaneous melanocytic tumor characterized by CRTC1-TRIM11 fusion gene. The descriptions on this paper are correctly based on the previous papers. The beautiful figures and well-organized tables on the current form will help readers understand the contents easily and precisely. This paper will be of interest to readers of this journal. I have only one concern mentioned below.

Author response: Thank you for your comments.

  • I am interested in macroscopic appearance of this tumor because it is one of the most important findings when clinicians inspect the skin of a patient. Are there any characteristic gross findings with this tumor? The authors are encouraged to add this information in Clinical features.

Author response: The gross findings were not included in the relative publications. Clinically it is only referred as a subcutaneous nodule. We have also included the clinical differential diagnosis.